# Targeting enhanced digestibility: Prioritizing low pith lignification to complement low p-coumaric acid content as environmental stress intensity increases

Oscar Main [1,2], Ana López-Malvar[1,3], Florence Meunier[4], Sophie Guillaume[1], Marie-Pierre Jacquemot[1], Paul-Louis Lopez-Marnet[1], Charlène Barotin[5], Anne Marmagne[1], Laurent Cézard[1], Sébastien Fargier[4], Sébastien Rey[4], Pierre Larsonneau[6], Nathalie Mangel[7], Anthony Uijttewaal[8], Matthieu Reymond[1], Sylvie Coursol [1], Valérie Méchin [1,9,10]*

**1** Université Paris Saclay, INRAE, AgroParisTech, Institute Jean-Pierre Bourgin for Plant Sciences (IJPB), Versailles, France, **2** IATE, University of Montpellier, INRAE, Institut Agro, Montpellier, France, **3** Facultad de Biología, Departamento de Biología Vegetal y Ciencias del Suelo, Agrobiología Ambiental, Calidad de Suelos y Plantas, Universidad de Vigo, As Lagoas Marcosende, Unidad Asociada a la Misión Biológica de Galicia (CSIC), Vigo, Spain, **4** Unité Expérimentale DiaScope, INRAE, Mauguio, France, **5** Unité de Recherche Pluridisciplinaire, Prairies, Plantes Fourragères, INRAE, Lusignan, France, **6** ARVALIS – Institut du Végétal, Station Expérimentale, Baziège, France, **7** ARVALIS – Institut du Végétal, Station Expérimentale, Boigneville, France, **8** ARVALIS – Institut du Végétal, Station Expérimentale de La Jaillière, Loireauxence, France, **9** INRAE, UMR AGAP Institute, Montpellier, France, **10** UMR AGAP Institute, University of Montpellier, CIRAD, INRAE, Institut Agro, Montpellier, France

\* valerie.mechin@inrae.fr

## Abstract

Forage maize is a central pillar of dairy cow feeding in France, directly influencing milk production. Drought significantly affects both maize yield and digestibility, which are both key parameters required for hybrid registration purposes. Research on maize inbred lines has revealed droughts' notable effect on dry matter and cell wall digestibilities due to changes in cell wall composition, directly impacting forage quality. No such studies have been performed on forage maize hybrids however, which are the main seed type used in the agricultural sector. In this paper, we aimed to understand the impact of water and heat stress on forage maize digestibility, and to uncover the factors controlling it. We grew a panel of eleven modern forage maize hybrids for two years under four different controlled water stress modalities. These plants were agronomically, biochemically and histologically assayed, allowing us to perform a multiscale analysis to determine the traits responsible for variations in digestibility. By establishing a comprehensive heat and water stress index, we classified the environmental conditions. We demonstrated that under severe stress, ear production decreases significantly, but dry matter digestibility can be maintained through increased cell wall digestibility. This boost in cell wall digestibility was due to a reduction in *p*-hydroxycinnamic acid content and changes in lignin distribution,

**Data availability statement:** The data underlying the results presented in the study are available from the Zenodo repository https://zenodo.org/records/17199793.

**Funding:** This work was entirely funded by the Plant2Pro Carnot institute in the frame of the MAMMA MIA project. Plant2Pro is supported by ANR (agreement #20 CARN 0024 01). This work has benefited from the support of IJPB's Plant Observatory platforms PO-Cyto (imaging and microscopy plateform). The IJPB laboratory also benefits from the support of the Saclay Plant Sciences-SPS (ANR-17-EUR-0007). The funders had no role in study design, data collection and analysis, decision to publish, or preparation of the manuscript. There was no additional external funding received for this study.

**Competing interests:** The authors have declared that no competing interests exist.

**Abbreviations:** ASI, Anthesis-silking interval; CWR, cell wall residue; BT, bundle tissues; DM, dry matter; DRT, dark rind tissues; ET, epidermis tissues; ETPP, estimated Penman evapotranspiration; EU, Experimental Unit; Feest, esterified ferulic acid content; Feeth, etherified ferulic acid content; G subunit, guaiacyl lignin subunit; H subunit, $p$-hydroxyphenyl lignin subunit; %ICS, percentage of the internode cross section; LRT, light rind tissues; MAT, Total protein content; MGDD, modified growing degree days; MT, medullary tissues; NIRS, near infrared spectrometry; Pflo, Period 1, from the 1st to the 15th of July; Ptot, Period 2, from the 16th of May to the 15th of August; PAR, photosynthetic active radiation; PCest, esterified $p$-coumaric acid content; WW, well-watered condition in Mauguio; WD1, moderate water-deficit condition in Mauguio; WD, severe water-deficit condition in Mauguio; RW, rainfed condition in Versailles; S subunit, syringyl lignin subunit; %S-PC, percentage of $p$-coumaroylated S monolignol units; SID, stress index; UFL, unité fourragère laitière (milk feed unit).

while lignin content and structure remained stable. The significant impact of lignin distribution on cell wall digestibility increased with the severity of the stress, reaching an extreme threshold where biochemical parameters solely account for digestibility variations. To improve maize digestibility, it will be necessary to better understand how the reduction in carbon flux under stress affects $p$-hydroxycinnamic acid levels without greatly impacting lignin content. Finally, our work suggests that the inclusion of moderate stress conditions in future maize breeding programs will be necessary to better adapt forage maize hybrids to climate change.

## 1. Introduction

Maize silage plays a crucial role in feeding livestock, particularly in large-scale dairy operations. Its high energy content and balanced nutritional profile make it an essential feed ingredient that supports optimal growth, milk production and overall cattle health. In the French dairy sector, forage maize hybrids are evaluated through their energetic value mainly driven by dry matter (DM) digestibility, which can be used to estimate milk production [1,2]. Previous studies have highlighted that this DM digestibility is controlled by the plant's Cell Wall (CW) digestibility and content, the latter-two factors being correlated [3–7]. The inclusion of DM digestibility as an official hybrid registration criterion in the late 1990s has helped to maintain and even increase overall forage maize quality in France [7].

Due to climate change, the periodicity, intensity and duration of extreme climatic events have all increased over recent years. These increases have also been observed in the main crop-growing regions [8–11]. These events have had clear visible impacts on overall crop yield levels [12,13]. Maize, as a summer plant, is sensitive to water stress, particularly during its flowering and subsequent grain-filling growth stages between July and August [14,15]. These periods correspond to when the crop requires significant quantities of water [16] to guarantee both pollination and correct grain yield [17–19]. Heat stress over these two growth stages can also exacerbate issues, jeopardizing pollen survival and seed setting and reducing potential grain yield [20–22]. Such drought conditions in the United States have had a significant impact on maize silage production. In 2023, maize production experienced a reduction of approximately 20% compared to the previous year, attributed to these challenging climatic conditions. In France, drought conditions have been found to reduce forage maize yield from 10 to 50% [23–25].

While this sensitivity to these environmental stresses is well known, work on selecting maize for these conditions has remained firmly rooted in efforts for the grain fraction. Yield increases obtained in past forage maize selection programs have also been based on higher grain yield [26,27] or greater tolerance to higher planting density [28]. A significant part of these higher yields may also be completely unrelated to selection efforts, and more due to warmer climates [29]. Comparable results regarding these yield questions are also found in water deficit conditions [30–33].

It is noteworthy that the maize CW is also affected by water deficit [6,34,35]. The CW is composed of structural elements such as polysaccharidic (cellulose and hemicelluloses) and phenolic compounds (lignin and *p*-hydroxycinnamic acids, *i.e.,* esterified *p*-coumaric acid or PCest, and esterified and etherified ferulic acids, or Feest and Feeth) crucial for providing support against the external environment of the plant [36,37]. These components are primarily deposited before the flowering period [38–40]. Previous work has identified a decrease in the content of structural components during water deficit over this period, but also a change in lignin and other components distribution between rind and pith of the stem, benefitting CW digestibility in maize [6,35,41–43]. Histological staining such as FASGA (Fucsina, Alcian blue, Safranina, Glicerina and Aqua) staining can allow the analysis of lignin distribution across these two stem regions [3,6,34,44]. The integration of qualitative traits alongside yield targets could provide essential breeding targets to improve overall forage quality under deficit conditions.

Previous selection efforts to improve forage maize digestibility have mainly concentrated on reducing and stabilising lignin content (personal communications with breeding partners). These efforts have been successful, as the lignin content levels observed in modern hybrids are extremely low and invariable, even under water deficit conditions [45]. As a potential source of improvement for future selection programs, lignin content appears to possess little room for further improvement. Therefore, in this paper, we investigated the effects of four contrasted growing conditions repeated across two years, including three different irrigation levels, on a set of modern early-flowering forage maize hybrids representative of the current French market. Our main objectives were (1) to develop a comprehensive environmental stress index (SID) to compare, to classify and to jointly analyse the different field environmental conditions; (2) to study the response of digestible yield in maize to increasing SID levels; (3) to reveal the main biochemical and histological parameters responsible for the variations in digestibility depending on stress intensity and to identify the parameters which could be used as an alternative to lignin as a potential selection trait.

## 2. Materials and methods

### 2.1. Stress management and field trials

Plants were grown at two environmentally contrasted INRAE experimental units (EU) in France across 2021 and 2022: Versailles-Saclay EU (Versailles, North of France, GPS: N: 48'48"33.391/ E: 2'5"7.237) and DiaScope EU (Mauguio, South of France, GPS: N: 43'36"52.438/ E: 3'58"34.419), hereafter referred to respectively as "Versailles" and "Mauguio". In both years, seeds were sown, and fields were managed uniformly to achieve a target planting density of 100,000 plants/ha. The experiment followed an incomplete Latin square design, with each variety planted in two rows across three blocks per environmental condition. In 2022, three hybrids were repeated across a further three blocks. A rainfed modality was repeated each year at Versailles (RW). Overall evapotranspiration and rainfall were followed daily at this location through a weather station located next to the field. This data was integrated into the Irré-Lis® irrigation management tool [46,47], which dictated whether irrigation was necessary or not to avoid water deficit. When this monitoring indicated a potential for water stress, a drip-fed irrigation system was used to compensate for the lack of rainfall with a 20 mm irrigation dose. Three different irrigation modalities were conducted each year at Mauguio: a well-watered modality (WW) with a triweekly 20 mm irrigation dose; a moderate water-deficit modality (WD1) which received a 15 mm irrigation dose upon soil water tension reaching –125 kPa; and a severe water-deficit modality (WD) which received a 13 mm irrigation dose upon soil water tension reaching –300 kPa. These two stress modalities were applied from 25 days post-sowing. These 3 watering scenarios were separated by a 20 m border between modalities to avoid irrigation overlap. Stress levels were monitored at Mauguio using three pairs of two Watermark® tensiometers present in each block per condition under a reference hybrid, at 30 and 60 cm depth in the soil, with the mean values used to trigger irrigation. These were installed alongside a pluviometer in each block. Slight differences affected both field trial years, leading to subtleties depending on the considered locations and conditions, as described below.

- **Versailles:** In 2021, the Versailles field trial received nearly 360 mm of total rainfall across the growing period, but in 2022 this quantity decreased 52% to 171 mm. This decrease required an irrigation input of 232 mm in 2022, bringing total water input up to 403 mm, which was not required in 2021. Mean temperatures also increased across both years, increasing from 16.8 to 18.5 °C from 2021 to 2022 (S1 Table).

- **Mauguio:** Rainfall did not significantly change across both years in Mauguio (113.5 mm in 2021, 116 mm in 2022), though the average temperature did increase from 22.2 to 23.8 °C from 2021 to 2022. In our two stressed conditions in 2021, irrigation was typically launched well before the tensiometer thresholds were reached as visual plant observations were more trusted. In 2022, we placed more trust in tensiometry data to dictate irrigation scheduling, which resulted in a reduced total irrigation dose for the three irrigation modalities in 2022 (13.71% for WW, 17.78% for WD1, 4.99% for WD) (S1 Table).

Significant annual environmental differences were observed over both growing years: 2021 presented a much milder and colder summer than 2022. These differences led us to consider the different yearly irrigation modalities as separate environmental conditions. As such, in our study we distinguished a total of eight environmental conditions: four in 2021 (RW 2021, WW 2021, WD1 2021 and WD 2021), and four in 2022 (RW 2022, WW 2022, WD1 2022 and WD 2022).

The hybrid material was selected from hybrids under evaluation in the S0-S1 Post Registration Evaluation Network, a national collaboration between the French agricultural technical institute ARVALIS and the maize and sorghum section of the UFS [48]. The 11 hybrids were selected according to three criteria: (i) maximising the yield to digestibility variability with a reduced flowering window; (ii) accurately representing the current state of the S0/S1 forage maize market; (iii) including diverse genetic and seed company diversity (S2 Table).

## 2.2. Agronomic analysis and sample harvest

Days to anthesis (noted when at least 50% of the plants in a block had pollen anthers that were at least two-thirds open) and days to silking (noted when at least 50% of the plants in a block had visible silks), were recorded and the anthesis-silking interval (ASI) was calculated. When the 32–34% DM harvest indicator was reached (evaluated by the milky to pasty grain transition), four homogenous representative plants were harvested per block (two in each row) per hybrid per condition at the ground level. The total plant and ear insertion height alongside the fresh weight were then measured. After grinding the plants in a Viking® GE 355, a representative sample of approximately 400 g was retained per block for drying in a forced-air oven at 55°C over 72h. The dried sample determined the block %DM content, allowing yield determination. It was then ground with a hammer mill (1 mm grid) for use in biochemical analysis. Additionally, for histological analysis, the internode below the main ear was harvested from three representative plants per block. This was realized simultaneously during the whole-plant harvests, for each hybrid in each block in all conditions and locations throughout both years. The resulting internode was then stored in 70% ethanol.

## 2.3. Stress index and environmental analysis

The SID was developed iteratively considering the effects of both water and heat stress as evenly as possible and on a daily basis. In this endeavour, the following equation is proposed:

$$SID = \left( \left( \frac{soil\ water\ deficit\ (mm)}{easily\ accessible\ water\ reserve\ (mm)} \right) * 100 \right) + \left( \frac{\left( \left( 24 * \left( \frac{n^o of\ hours\ spent > 30^o C}{24} \right) + 1 \right) * \left( \frac{maximum\ temperature\ of\ the\ day (^o C)}{heat\ stress\ threshold (^o C)} \right) \right)}{2400 + 1} \right)$$

The water stress aspect considers the percentage of utilisation of the available water reserve. This is usually calculated as a percentage, wherein if the percentages exceed 100%, the plants have theoretically used up all the available water

and have entered a state of water stress. This value can be measured through different methods. In this paper, we calculated this part of the equation by dividing the soil water deficit by the easily accessible water reserve, both in mm, obtained using the ARVALIS Irré-LIS® (irrelis.arvalisinstitutduvegetal.fr) [46,47] irrigation decision-making tool. The heat aspect estimates the impact of heat stress on plant development. This aspect is split into two parts, the first quantifying the duration of time spent above the specified heat stress threshold (30°C for maize), and the second considering the extent to which the maximum temperature reached during the day exceeded the heat stress threshold. The SID values were calculated according to two key periods:

- 1st to 15th of July: this period frames both the male and female flowering periods. It was referred to as Pflo, for Period over flowering.

- 16th of May to 15th of August: this period covers the maize development and covers the potential months of drought in France. It excluded stages more related to periods affecting the number of plants and desiccation stages and focused on the overall plant yield. This period was referred to as Ptot, for Period over total culture.

For statistical analysis, data from these time periods are considered in semi-monthly time windows instead of weekly periods. This decision was made to achieve sufficient periodic precision while avoiding analytical overloading. SID values are calculated per day and were then summed over the considered periods. Additional environmental traits were acquired from weather stations installed at both experimental units, connected via the Climatik environmental INRAE database organised by the INRAE AgroClim unit. These included:

- The sum of the Total Water Input (TWI, combining both total rainfall and irrigation, when applicable, in mm) subtracted by the Penman Predicted Evapotranspiration (ETPP), abbreviated as TWI-ETPP and measured either as a daily mean or a sum for both periods across both years. These values are typically negative, but for calculation purposes are inverted when necessary.

- The daily mean temperature (in °C) for both time periods.

- The mean daily air humidity (in %) for both time periods.

- The sum of the Photosynthetically Active Radiation, abbreviated to PAR (Joules per cm²) for both time periods.

- The Heat Stress Duration, abbreviated to HSD (number of hours spent above 30°C) for both time periods.

## 2.4.  Biochemical analysis

For DM samples, DM digestibility was measured in duplicate for the 11 hybrids, 3 blocks and 8 growing conditions (132 samples in 2021, 168 in 2022) using the method described in [49]. Digestible yield was then calculated on these samples (digestible yield = yield * (DM digestibility/100); [45]). A 66 samples subset from the two studied years, limited to conditions from Mauguio was also assayed for soluble sugar content at the INRAE Lusignan site (33 per year).

Following block-level analysis using both agronomic and biochemical data to identify the presence of any block effects in the different locations and conditions, two representative blocks were retained per condition and per year for CW residue (CWR) extraction [50]. A total of 88 samples were extracted in 2021 and 112 in 2022. CW digestibility was then measured for all extracted samples for both years (200 in total) according to Lopez-Marnet *et al.*, 2021 [49].

CW samples were also assayed for lignin content by subjecting approximately 7–8 mg of extracted CW to an acetyl bromide treatment as presented in Fukushima and Hatfield, 2001 [51]. Lignin structure and composition were determined through thiacidolysis by treating 13–15 mg of extracted CW with boron trifluoride and ethanol heated at 100°C for 4 h (oil bath). Following this, several elements are dosed by gas chomatrography (GC). These include the different monolignol units (H, G, S), alongside the βO4 yield. The latter trait represents the quantity of the ether bond present between lignin

monolignol units. The greater presence of this type of bonds usually indicates the presence of a linear lignin, which is resist enzymatic degradation less than highly cross-linked and condensed lignin [52]. This method also allows the measurment of the percentage of S monolignol units which are p-coumaroylated (%S-PC). Given the large number of samples and the throughput of the thioacidolysis assay, only one block per condition per year was assayed for thiacidolysis (44 per year). Additionally, 50 mg of CWR per sample (2 blocks for 2021, 88 samples; 1 block for 2022 due to time constraints, 56 samples) was submitted to alkaline hydrolysis under NaOH 2N at ambient temperature for 20 h to allow liberation of acids from ester links [3] and alkaline hydrolysis under NaOH 4N at 170°C to allow liberation of both esterified and etherified acids. Following extraction [53], esterified p-coumaric acid (PCest), esterified ferulic acid (Feest) and etherified ferulic Feeth) were then analysed in high-performance liquid chromatography with ultraviolet detection (HPLC-UV).

### 2.5. Histological analysis

**2.5.1. Image acquisition and treatment.** Two of the three harvested internodes located below the main ear were sampled at 1.5 cm under the upper node and cross-sections with thicknesses of 150 μm were obtained with a GSL1 sledge-microtome [54]. These were then stained under agitation as described by El Hage *et al.*, 2018 [6], wherein internodes are stained for 24 h in FASGA (1:8, v/v), washed for another 24 h in distilled water before mounting between slide and coverslip. Images were then immediately acquired through an AxioImager Z2 Zeiss, as described in Legland *et al.*, 2017 [55]. During this staining process, lignified tissues become red while non-lignified tissues become blue. Cross-section images were obtained in a Metafer automated slide scanner platform (MetaSystems, GmbH, Altlussheim, Germany). Images were obtained with a 5x objective lens and then reduced in size *via* the MetaViewer software to obtain a final resolution of 5.17 μm per pixel.

**2.5.2. Image segmentation data treatment and analysis.** Image handling and modifications were performed using the ImageJ Fiji version [56,57]. Images were automatically segmented utilising the ImageJ plugin developed by Lopez-Marnet *et al.*, 2022 [58]. In summary, this plugin, originally established for segmenting FASGA-stained maize internodes, segments 44 different zones (alongside a few summary variables) in raw pixel amounts. The respective pixel surface area is then divided by the total pixel size of the cross-section, allowing the calculation of each tissue type in a percentage of the total cross-section surface (%ICS, for % of the internode cross-section). The plugin quantifies different types of tissues depending on their localisation and colour. These include two types of rind tissues (Dark Rind Tissues, referred to as DRT; Light Rind Tissues, referred to as LRT), different medullary tissues (MT) in the pith, and several tissues linked either to the epidermis (ET) or to tissues close to or included within the vascular bundle (BT). The plugin also calculates sums of these tissues, alongside the total overall lignified and low lignified surfaces.

Separately, these tissues can contribute significantly to the identification of histological differences between environmental conditions. However, their important number can complexify analysis. To go further, we have used our expertise to group segmented tissues together to give these traits a stronger biological meaning. Statistical analyses have been performed on both types of data and are much more readable when performed on synthetic traits. The synthetic traits have been created according to their Fasga color (red or blue) in the considered tissue and their level of digestibility according to Lopez-Marnet *et al.*, 2022 [58]:

• Red or blue tissues:

  ◦ Blue pith tissues (MT2, MT5).

  ◦ Blue dark rind tissues (DRT1, DRT3, DRT6).

  ◦ Blue light rind tissues (LRT1, LRT3, LRT6).

  ◦ Blue bundle tissues (BT2, BT5).

◦ Red dark rind tissues (DRT2, DRT4, DRT5, DRT7, DRT8, DRT9).

◦ Red light rind tissues (LRT2, LRT4, LRT5, LRT7, LRT8, LRT9).

◦ Red bundle tissues (BT1, BT3, BT4).

- Digestible or indigestible tissues in medullary region:

◦ Highly digestible tissues (MT2b, MT5b).

◦ Digestible tissues (MT2a, MT5a).

◦ Highly indigestible tissues (MT1a, MT4a).

◦ Indigestible tissues (MT1b, MT4b).

## 2.6. Statistical analysis

Data was statistically analysed using R 4.3 with the RStudio interface [59,60]. Data was averaged across all blocks per hybrid for each condition per year. Data management was performed using tidyverse and dplyr packages [61,62]. Data normality was verified *via* Shapiro-Wilk testing and residual dispersion analysis.

ANOVA analysis was performed utilising a linear model accounting for hybrid and condition effects. Additional analysis was also performed on the row and column field trial design effects for all conditions to identify any potential position effect within the field. If these effects were identified, they were included within the ANOVA model as covariables, otherwise the analysis was performed utilising a simpler model:

$$Y_{ijkl} = \mu + H_i + C_j + (HC)_{ij} + c_{jk} + r_1 + E_{ijkl}$$

Where $Y_{ijkl}$ is the value for the given trait of the *i*th Hybrid in the *j*th Condition localised, if necessary if the *k*th column and the *l*th row within the field, with μ as the intercept. The decision to retain or not the additive model was calculated through an F test for interaction as described in Main *et al*., 2023 [45]. As the environmental analysis indicated annual differences across both growth sites, the conditions were examined on a year-by-year basis by pairing each modality with its respective year (e.g., "RW_2021", "RW_2022").

Bar plots and dot plots were generated using the ggplot library [63]. When data points were sufficiently available, error bars were added by calculating the standard error values for each variable. Principal component analysis (PCA) was performed using the factoextra package [64], with analysis ran on the first six dimensions.

Linear correlations were calculated using ggpbur [65]. Pearson correlations with their associated p-values were calculated utilising the ggcor and corrplot packages [66,67]. Linear multiple regression models were built utilising the olsrr package [68]. In some model-generating runs, some variables may be added to the model without significantly increasing the model precision ($R^2$) or reducing the Root Mean Square Error (RMSE). To avoid these issues, and to therefore simplify the analysis, models were cut off when variable addition did not improve model precision by at least 4% or if not RMSE was not notably decreased. Regression analysis was performed on a per environment basis. Variables retained for analysis in the regression models are present in S3 Table.

All information that was included or considered within this publication are publicly accessible through the Main, 2025 upload [69].

## 3. Results

### 3.1. A stress index integrating soil water deficit and air temperature for environmental stress ranking

For each growing condition, we collected standard agroclimatic data across the two considered phases (Pflo and Ptot). A PCA was performed for all agroclimatic traits studied under the eight conditions. The first two dimensions of the PCA

explained 93.9% of the variation of all agroclimatic traits (Fig 1A). Dimension 1 explained 65.5% of the variation and separated the year effect, with 2021 conditions being influenced by air humidity, while those in 2022 were influenced by temperature and PAR. Interestingly, the year effect for 2022 appeared to repeat the position of 2021 but pulled towards the right-hand side. Dimension 2 explained 28.4% of the variability and separated the stress intensities, with the WD conditions leading alongside plant water rain and irrigation minus ETPP (86.64%). When considered as illustrative variables, the SID sums calculated over Pflo and Ptot phases were between water stress (rain & irrigation minus ETPP) and heat

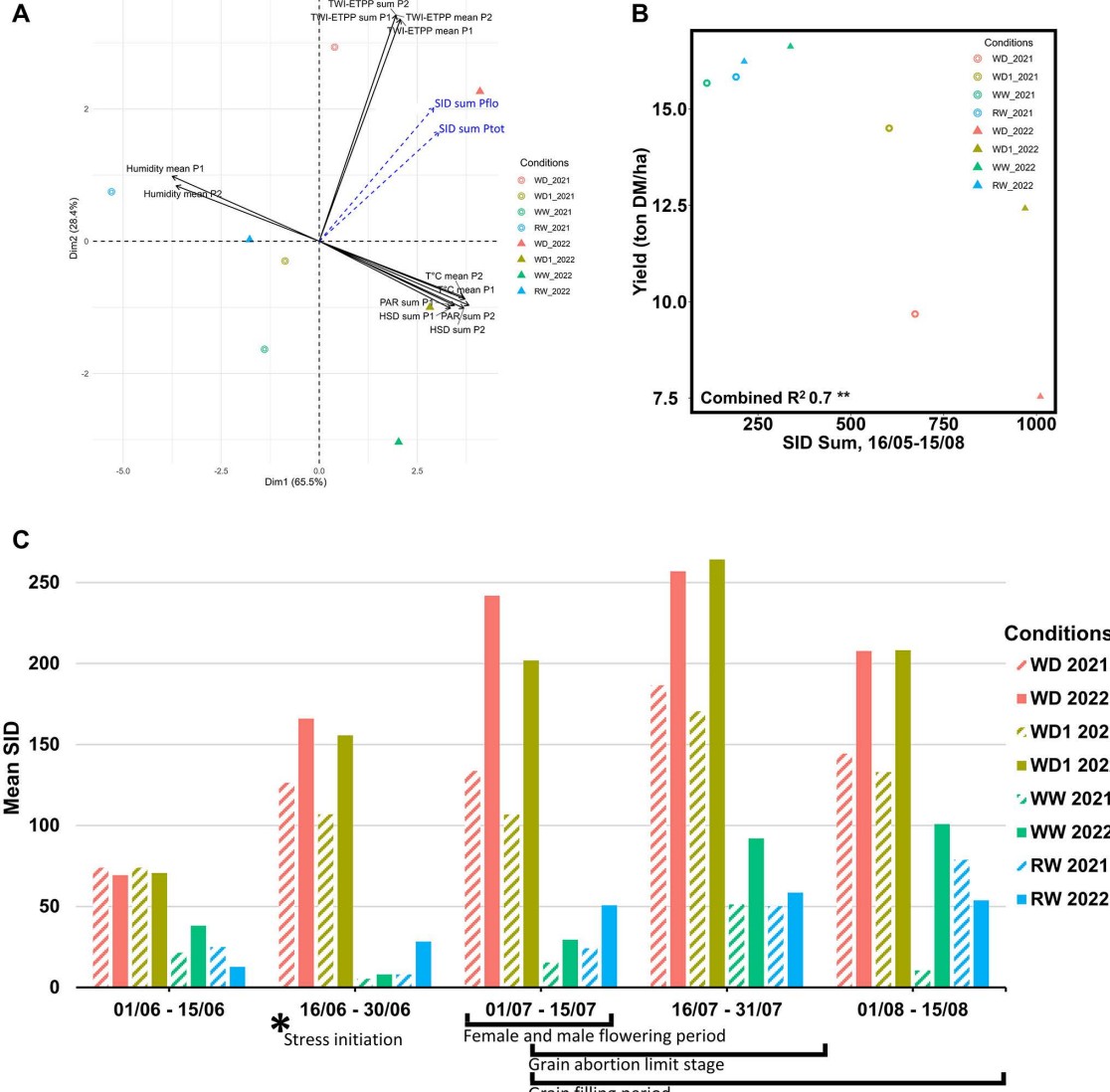

**Fig 1. Analysis of climatic, environmental and stress index data across the main growth periods.** (A) Distribution of the main environmental and climatic variables, alongside SID variables projected as spectator variables on the first two principal components. (B) Linear correlation between the SID Sum between the 16th of May to the 15th of August to overall whole plant yield in ton DM/ha at harvest. (C) Mean SID values per half month period from the 1st of June to the 15th of August for the eight different conditions. Main physiological stages, including stress initiation and both flowering and grain-related periods are added below.

stress (Temperature, PAR) and tended to be closer to the water deficit traits. A significant negative correlation was also observed between SID values and plant yield (Fig 1B), and a significant positive correlation between SID values and ASI.

SID values were then averaged per half-month period from the 1st of June to the 15th of August across all conditions and both years, with these periods covering the most important developmental stages of maize growth, which also correspond to the times of greatest stress sensitivity. The SID accurately quantified the condition differences for both water stress conditions (WD1 and WD) in 2021 and 2022, indicating a progressive increase in stress between the irrigated, moderate, and severe stress conditions during anthesis and silking (Fig 1C). While the SID values in the WW and RW conditions did not exceed the 100% thresholds, an annual effect was observed.

Interestingly, while the irrigation doses between both studied years were similar between both sets of conditions, the observed SID values varied greatly within conditions. For example, the SID values obtained for WD1 2022 almost resemble those present in WD 2021, while the values observed for WD 2022 surpass all other measurements. These differences led us to group the different conditions depending on the observed SID results. The irrigated and rainfed conditions were grouped into a "humid conditions" category, which represented the lowest stress intensity values. Then, we decided to create a "moderate to severe stress" category to represent conditions wherein a stress was present, but with varying levels of intensity. This contained, in order of increasing stress, WD1 2021, WD 2021 and WD1 2022. This was then followed with a "severe stress" category, containing WD 2022.

### 3.2. A strong impact of stress intensity levels on the main qualitative parameters

Key agronomic and qualitative criteria were obtained from all hybrids across the growing conditions to accurately assess the effects of the different stress levels.

In forage maize hybrid evaluation networks, the two most important criterion for evaluation are yield and DM digestibility. Therefore, in 2021 and 2022, we studied the relationship between these two major traits (Fig 2A and 2B). In 2021, the observed results were as expected and similar to those found in previous projects: DM digestibility increased in the stressed conditions, and this proportionally to the decrease in yield. However, results found in 2022 inversed this trend: while the humid and moderately stressed conditions do not differ in results across both years, DM digestibility significantly decreased in WD 2022 (Table 1, Fig 2B). This annual impact is also observed at the hybrid-level. In 2021, hybrids J and E emerged for their resilience to stress conditions (hybrid J in WD 2021 was grouped with WD1 2021 individuals, while the same hybrid in WD1 2021 was grouped with those in WW 2021), or their overall performance under moderate stress (hybrid E yielded as much in WD1 2021 as in WW 2021) (Fig 2A). Such distinctions were absent in 2022, with hybrids firmly remaining inside their condition group (Fig 2B).

Digestible yield values were obtained for all conditions through the combination of both yield and DM digestibility values (Table 1, Fig 2C). Throughout the humid conditions, mean digestible yield levels peaked in WW 2022 with 12.73 tons of digestible DM/ha, which remained similar to levels found in WW 2021 (11.75 tons of digestible DM/ha). These values were only slightly higher than those found in RW 2021 and 2022, where we observed 11.47 tons of digestible DM/ha in both years.

In the moderate to severe stress conditions, the reduction in yield can be compensated by an increase in DM digestibility as long as the yield drop is not too important. As such, in WD1 2021 we identified digestible yield levels which were not significantly different from those observed in the RW 2021 and RW 2022 conditions, though this level was reduced by 7.14% (10.65 *vs* 11.47 tons of digestible DM/ha). Results from WD1 2022 follow this trend with 8.39 tons of digestible DM/ha, reflecting a 21.22% decline between the two years. However, this reduction in digestible yield was not significant when compared to WD1 2021, or both years in the RW conditions. In contrast, the digestible yield observed in WD 2021, with a mean digestible yield of 7.67 tons of digestible DM/ha, was significantly different from the two others moderate to severe stressed conditions. Additionally, a significant difference was observed under the severe stress condition of WD 2022. The drop in DM digestibility and yield in this condition resulted in a digestible yield of 4.48 tons of digestible DM/ha, which was significantly different from all other conditions.

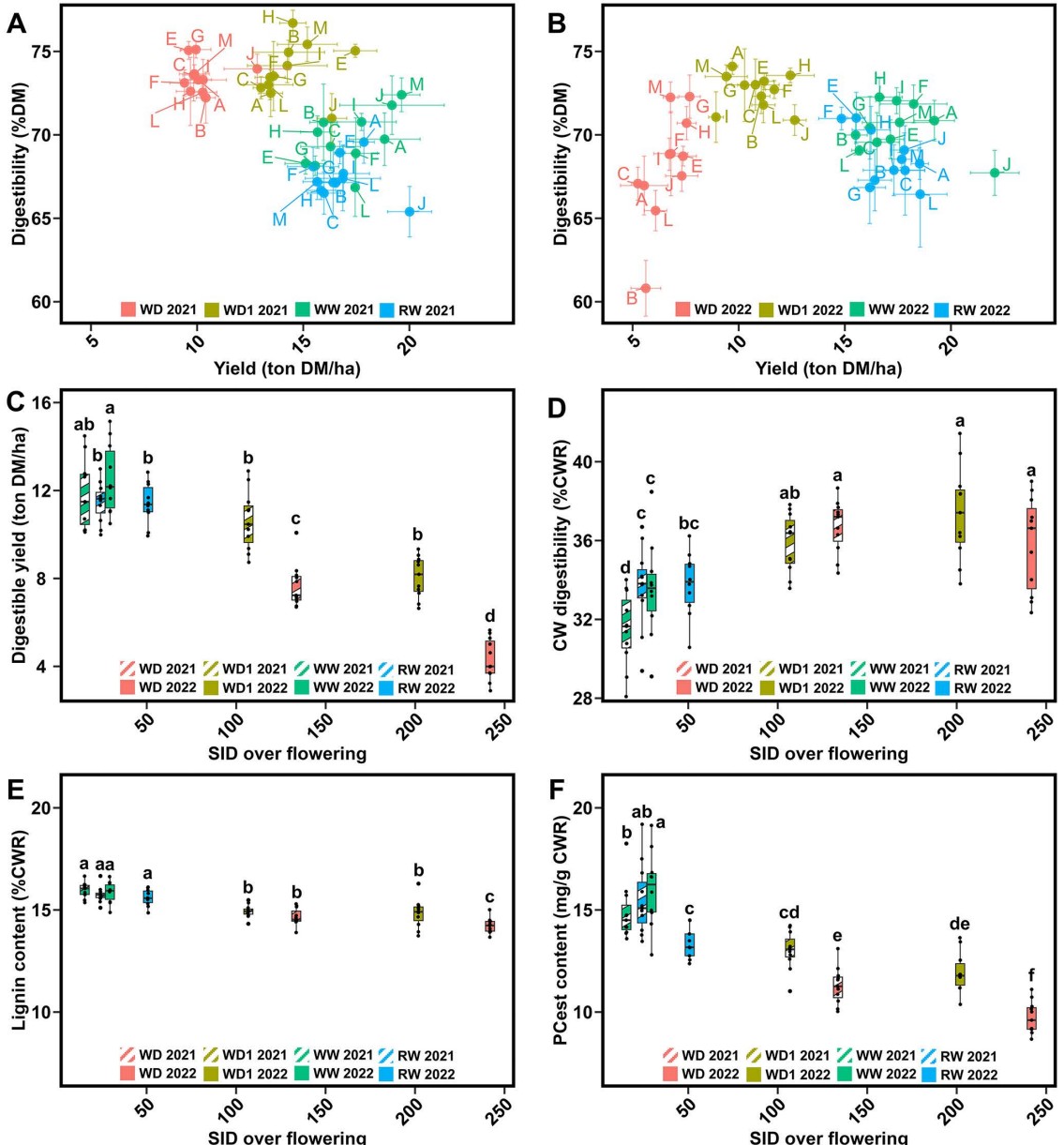

**Fig 2. Impact of stress intensity across 2021 and 2022 on key agronomic and biochemical markers.** (A) Impact on the DM digestibility to yield relationship in 2021 (A) and 2022 (B), with each dot equal to the mean values of the three blocks for the given hybrid, with standard error bars represented. The stress intensity effect is then observed for digestible yield (C), CW digestibility (D), lignin content (E) and PCest content (F), for all eight growing conditions. Bars represent the means and are hashed in conditions from 2021 and in full colours for those from 2022, and the error bars indicate the standard error margins. Letters on top of bars represent the Tukey class for the given condition ($P < 0.05$).

Several compositional traits in forage maize are often studied to explain variations in both yield and DM digestibility-related traits. One such trait, soluble sugar content, did not significantly explain any variation. Starch content, which can often approximate grain yield, showed a significant correlation with DM digestibility (overall $R^2 = 63\%$ in 2021, 16% in 2022). As stress intensity increased, the decrease in starch content was observed, which was associated with the reduced

**Table 1. Impact of the environmental conditions on the main agronomic and biochemical traits studied. Mean values are indicated. Different letters for the same trait correspond to statistically significant differences between conditions (*P*<0.05). *na*: not available.**

| Traits | RW 2021 | RW 2022 | WW 2021 | WW 2022 | WD1 2021 | WD1 2022 | WD 2021 | WD 2022 |
|---|---|---|---|---|---|---|---|---|
| Yield (ton DM/ha) | 16.95[ab] | 16.66[b] | 16.86[ab] | 18.24[a] | 14.40[c] | 11.57[d] | 10.40[d] | 6.60[e] |
| Whole ear proportion in yield (%) | *na* | 40.64[a] | *na* | 34.86[b] | *na* | 30.79[c] | *na* | 19.60[d] |
| DM digestibility (%DM) | 67.60 cd | 68.76[bcd] | 69.59[bc] | 69.64[b] | 73.95[a] | 72.49[a] | 73.65[a] | 67.51[d] |
| Digestible yield (ton DM/ha) | 11.47[b] | 11.47[b] | 11.75[ab] | 12.73[a] | 10.65[b] | 8.39[b] | 7.67[c] | 4.48[d] |
| DM digestibility without hybrid B (%DM) | 67.66[c] | 68.82[bc] | 69.45[bc] | 69.68[b] | 73.91[a] | 72.45[a] | 73.63[a] | 68.11[b] |
| Cell Wall content (%DM) | 50.10[b] | 48.03[bc] | 46.27 cd | 44.5[de] | 42.95[e] | 44.68[de] | 45.02[de] | 52.80[a] |
| CW digestibility (%CWR) | 33.66[c] | 34.08[bc] | 31.37[d] | 33.86[c] | 35.94[ab] | 37.15[a] | 36.72[a] | 35.97[a] |
| Lignin content (%CWR) | 15.75[a] | 15.69[a] | 15.98[a] | 15.85[a] | 14.92[b] | 14.89[b] | 14.66[b] | 14.20[c] |
| PCest (mg/g CWR) | 15.51[ab] | 13.42[c] | 14.95[b] | 16.60[a] | 13.02 cd | 11.99[de] | 11.29[e] | 9.67[f] |
| Feest (mg/g CWR) | 8.53[a] | 7.26[bc] | 7.65[bc] | 7.04[c] | 7.97[ab] | 7.68[bc] | 7.38[bc] | 5.83[d] |
| Feeth (mg/g CWR) | 2.67[ab] | 3.37[a] | 2.05[bc] | 2.49[b] | 2.45[b] | 2.60[ab] | 1.66[c] | 2.53[b] |
| Fetot (mg/g CWR) | 11.12[a] | 10.64[ab] | 9.70[bcd] | 9.53 cd | 10.42[abc] | 10.29[abc] | 9.04[de] | 8.35[e] |
| H subunit (µmol/g lignin content) | 14.31[a] | 15.07[a] | 15.82[a] | 12.90[a] | 15.51[a] | 14.07[a] | 14.93[a] | 15.78[a] |
| G subunit (µmol/g lignin content) | 326.25[a] | 340.24[a] | 351.24[a] | 332.79[a] | 339.33[a] | 328.85[a] | 312.49[a] | 348.73[a] |
| S subunit (µmol/g lignin content) | 309.66[b] | 333.73[ab] | 334.03[ab] | 366.83[a] | 342.30[ab] | 341.21[ab] | 320.39[b] | 343.78[ab] |

production of ears (Table 1). Ears provide digestible starch, while stems and leaves contribute both a soluble digestible fraction and a non-soluble fraction consisting of CW traits and of non-total digestibility.

DM digestibility is the result of the overall digestibility of the combined stem, leaves and grain compartments. The similar levels of DM digestibility observed between WD1 in 2021 and 2022 are likely due to the maintenance of the ear-provided starch and of CW digestibility. This idea is concurred by findings that CW digestibility levels did not significantly differ among the four stressed conditions across both years (Fig 2D), though these levels were systemically higher than those found in humid conditions. Despite a significant reduction in ear proportion, DM digestibility in WD 2022 remained consistent with WW 2022. This is especially evident when considering DM digestibility without hybrid B, which exhibited the greatest drop in DM digestibility (Table 1). This drop was likely due to the increase in CW digestibility (Fig 2D), prompting further investigation into CW-related traits. Lignin content showed small variations between hybrids within a given environment (Table 1, Fig 2E). The same trend was observed between environments, even under WD conditions for both years. Approximately a 1.5-point difference was detected between WW and WD conditions in the same year, equating to a 10% variation and close to the measurement error when estimating lignin content (Table 1). On the other hand, esterified *p*-coumaric acid levels displayed a significant variability both within and between conditions (Fig 2F). Such levels of variation were also found for the other studied *p*-hydroxycinnamic acid levels (Table 1). We also observed a significant increase in the different number of Tukey groupings for PCest levels. While lignin content results returned a total of three different Tukey class levels, six were identified for PCest content. Once again, the WD 2022 condition presented the greatest PCest content drop compared to all other conditions, with a reduction of 36% when compared to the humid conditions and 20% when compared to the moderate to severely stress conditions (Table 1).

### 3.3. Agronomic, biochemical and histological traits significantly respond to both the genetic and environmental factors

When combined across years and conditions, the ANOVA trait analysis revealed a wide range of significant interactions between conditions and hybrid (S4 Table).

Significant interactions were identified for all agronomic traits and most biochemical traits, except the S/G subunit ratio and the β04 yield. Results were more varied for the original histological traits, with far more traits with no significant interactions between the hybrid and condition factors. Analysed separately, segmented regions from cross section image analysis rarely displayed significant effects or interactions. When combined according to their stained colour or known links to digestibility however, we systematically identified significant interactions between the hybrid and condition factors.

### 3.4. Histological tissues play different roles in relation to both CW digestibility and composition

We found that while lignin content was affected by water deficit intensity, these variations presented a very limited range. These invariable responses could be hiding changes in stem lignification distribution under stress. No visible changes in the proportion of the rind surface occupied by either dark (DRT) or light rind tissues (LRT) were observed until 2022 (S1 Fig). While tempting to explain the changes that only occurred in 2022 by a change in lignin content, the reductions identified in WD 2022 (Table 1) were not found to be significantly correlated to either the DRT or LRT surfaces. This difference between tissues was also noticeable when visually observing the output of the image segmentation plugin on hybrid H, our agronomic reference, across the 4 conditions of 2022 (bottom of Fig 3A–3D). In the rind, LRT area progressively decreased under stress conditions, inversely to the area of DRT. The opposite was observed under humid conditions. Effects were also observed in the pith (top of Fig 3A–3D). As stress levels increased, a FASGA stained blue low-lignified band progressively increased under the rind section. Under humid conditions, this band was decreased and intercepted before the rind by a layer of lignified tissues. We found that DRT traits were negatively correlated with CW digestibility, while LRT traits were positively correlated. This may be explained by CW composition, as we found that DRT was often positively related to lignin content, β04 yield, S subunit content, PCest content, Feest content, and %S-PC, which were in turn negatively correlated with CW digestibility. Conversely, LRT was often negatively correlated with some of the same traits. Interestingly, in the WD1 and WD 2022 conditions, LRT traits were positively correlated with lignin structures such as β04 yield and with PCest, while in these conditions, DRT was not significantly related to these biochemical traits.

### 3.5. Histological traits contribute progressively more to CW digestibility under moderate stress conditions than under humid conditions

To determine the factors responsible for CW digestibility variation under each growing condition, stepwise multiple linear regression models were run per condition and year using the factors presented in the Materials and Methods (Fig 4). For the humid conditions, we found that 97% of the CW digestibility variation was explained by the model in RW 2021. Of this, 89% was due to four biochemical traits (*p*-hydroxycinnamic acids, 67%; lignin structure, 22%) and 5% was due to one histological trait (digestible blue). In RW 2022, CW digestibility was only explained to 66%, with one biochemical trait (*p*-hydroxycinnamic acids) and one histological pith trait, with an $R^2$ of 23%. In WW 2021, 96% of CW digestibility was explained by three biochemical traits (lignin structure, 93%*; p*-hydroxycinnamic acids, 3%). In WW 2022, the model explained 94% of CW digestibility, of which 89% was explained by three biochemical traits (*p*-hydroxycinnamic acids, 82%; lignin structure, 7%), and 7% by one histological pith trait.

For the moderate stress conditions, beginning with the WD1 2021 condition, CW digestibility was explained up to 82% by the resulting model. Contrary to the previous results identified among the humid conditions, this model was mostly explained up to 50% by a single histological trait (highly digestible blue tissues), and then to 32% by three biochemical traits (*p*-hydroxycinnamic acids). This greater importance of histological traits continues into the next condition with WD 2021, where the model explained 77% of CW digestibility with three histological traits having explained 52% (rind tissues, 29%; highly digestible blue tissues, 23%) and one biochemical tissue at 25% (S subunit etherified by *p*-coumaric acid). The greater role of histological tissues is then slightly reduced in WD1 2022. Here, the

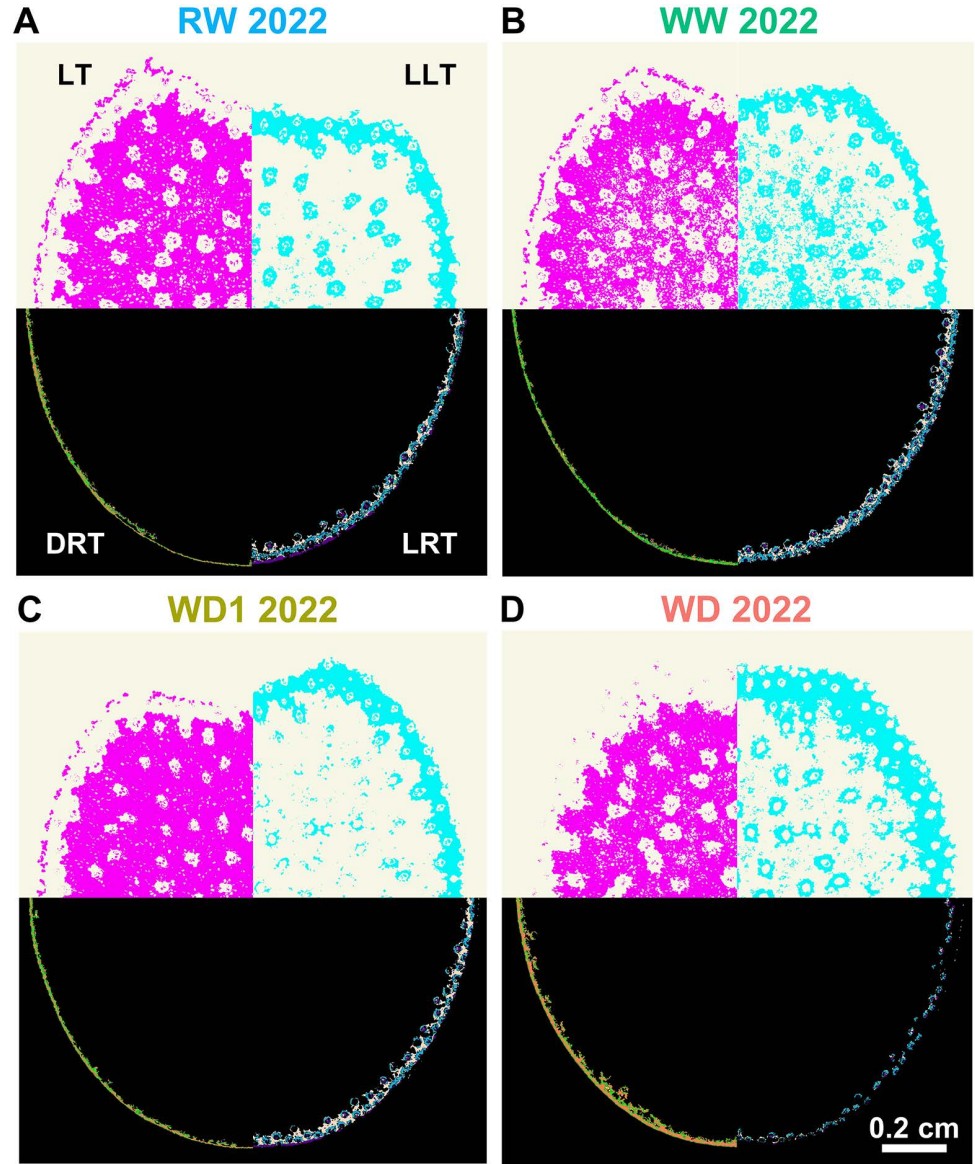

**Fig 3. Impact of water deficit intensity on tissular distribution of biochemical traits.** Distribution of the four main summarized tissue types in hybrid H for the humid (A, B), moderate (C) and severe (D) stress conditions. In each image, Lignified Tissues (LT) are presented in the top left quarter while Low Lignified Tissues (LLT) are opposite on the top right quarter. On the bottom left quarter, calculated Dark Rind Tissues (DRT) are placed opposite Light Rind Tissues (LRT) in the bottom right quarter. Tissue types are represented in (A) and image scale in (D). All images are set at the same scale.

model explained 88% of CW digestibility. 60% of this variation was due to two biochemical traits (S subunit esterified by *p*-coumaric acid, 53%; lignin structure, 6%) and 28% by two histological traits (highly digestible blue, 18%; rind tissues, 10%).

In the extremely severe stress condition, with WD 2022, the role of histological tissues is reverted to that previously observed among the humid conditions. In this condition, the model explains 95% of CW digestibility variation. 90% of this model is explained by three biochemical traits (lignin structure, 59%; *p*-hydroxycinnamic acids, 23%; S subunit esterified by *p*-coumaric acid, 8%) and one histological trait for the remaining 5% by one histological trait (highly digestible blue tissues).

| Group | Step | Introduced Trait | Trait Pearson correlation | Partial R² | Model R² | RMSE | Model Coefficient | B vs H |
|---|---|---|---|---|---|---|---|---|
| | | **Step Wise Selection for CW digestibility** | | | | | | |
| | | **RW 2021** | | | | | | |
| | 1 | PCest (mg/g CWR) | -0.735 | 0.61 | 0.63 | 1.31792 | -1.779401878 | |
| | 2 | S/G subunit ratio | ns | 0.16 | 0.79 | 0.98271 | 77.88190082 | |
| | 3 | Feeth (mg/g CWR) | -0.517 | 0.07 | 0.86 | 0.80285 | -0.867882046 | |
| | 4 | Digestible blue (%ICS) | -0.392 | 0.05 | 0.91 | 0.65247 | -0.041620147 | |
| | 5 | S subunit (µmol/g lignin content) | -0.614 | 0.06 | 0.97 | 0.36055 | -0.214060794 | |
| Humid conditions | | **RW 2022** | | | | | | |
| | 1 | PCest (mg/g CWR) | -0.696 | 0.42 | 0.42 | 1.3502 | 1.606019122 | |
| | 4 | Blue pith tissues (%ICS) | ns | 0.23 | 0.66 | 1.03895 | -0.053979407 | |
| | | **WW 2021** | | | | | | |
| | 1 | S subunit (µmol/g lignin content) | -0.906 | 0.82 | 0.82 | 0.8514 | -0.068696105 | 100% B |
| | 2 | H subunit (µmol/g lignin content) | ns | 0.11 | 0.93 | 0.5313 | 0.297870094 | |
| | 3 | Feest (mg/g CWR) | ns | 0.03 | 0.96 | 0.38795 | -0.435680097 | |
| | | **WW 2022** | | | | | | |
| | 1 | PCest (mg/g CWR) | -0.69 | 0.55 | 0.55 | 2.47668 | -1.640223117 | |
| | 2 | Feest (mg/g CWR) | 0.675 | 0.27 | 0.82 | 1.58286 | 2.942164253 | |
| | 3 | Blue pith tissues (%ICS) | ns | 0.05 | 0.87 | 1.33421 | -0.115575588 | |
| | 4 | β04 yield (µmol/g lignin content) | -0.491 | 0.07 | 0.94 | 0.87515 | 0.018631186 | |
| | | **WD1 2021** | | | | | | |
| | 1 | Highly digestible tissues (%ICS) | 0.509 | 0.50 | 0.50 | 1.01569 | 0.067375735 | |
| | 2 | PCest (mg/g CWR) | -0.606 | 0.19 | 0.69 | 0.79586 | -0.903905845 | |
| | 3 | Feest (mg/g CWR) | ns | 0.08 | 0.77 | 0.68851 | 0.350242709 | |
| | 4 | Feeth (mg/g CWR) | -0.324 | 0.05 | 0.82 | 0.60935 | -0.450646685 | |
| Moderate to severe stress conditions | | **WD 2021** | | | | | | |
| | 1 | Blue dark rind tissues (%ICS) | ns | 0.21 | 0.21 | 1.50221 | 0.502720745 | |
| | 2 | Highly digestible tissues (%ICS) | ns | 0.23 | 0.44 | 1.26504 | 0.113239805 | |
| | 3 | S-PC (%) | -0.377 | 0.25 | 0.69 | 0.94205 | -0.555336042 | |
| | 4 | Blue light rind tissues (%ICS) | ns | 0.08 | 0.77 | 0.81786 | -7.365954833 | |
| | | **WD1 2022** | | | | | | |
| | 1 | S-PC (%) | -0.73 | 0.53 | 0.53 | 1.79697 | -2.739803251 | |
| | 2 | Highly digestible tissues (%ICS) | ns | 0.18 | 0.71 | 1.40899 | 0.097192688 | |
| | 3 | Red dark rind tissues (%ICS) | ns | 0.10 | 0.81 | 1.13841 | 1.039782061 | |
| | 4 | H subunit (µmol/g lignin content) | ns | 0.06 | 0.88 | 0.9226 | 0.600063751 | |
| Severe stress condition | | **WD 2022** | | | | | | |
| | 1 | S/G subunit ratio | 0.769 | 0.59 | 0.59 | 1.9764 | 46.96861765 | |
| | 2 | PCest (mg/g CWR) | ns | 0.23 | 0.82 | 1.30 | -3.751702826 | |
| | 3 | S-PC (%) | -0.749 | 0.08 | 0.91 | 0.93648 | 2.070458323 | |
| | 4 | Highly digestible tissues (%ICS) | ns | 0.05 | 0.95 | 0.66679 | 0.047991574 | |

**Fig 4. Biochemical and histological trait relation to CW digestibility and regression model construction.** Conditions are listed progressively depending on their level of stress intensity. Introduced traits are listed depending on the step they were introduced into the model. Their respective direct relation to CW digestibility is then evaluated via Pearson correlation. If this Pearson correlation is only significant under $P<0,1$, the value is italicized. This is followed by their individual contribution to the regression model, and then the overall model R square. Following this is then the model root mean square error (RMSE) and it's evolution depending on the introduced traits, and then the model coefficient for each trait. A pie chart then represents the contribution of both biochemical (Biochemistry, B) versus histological traits (Histology, H) traits within the indicated model, with the total being equal to the sum of both parts.

## 4. Discussion

### 4.1. The simple but robust stress index provides a key environmental ranking tool

As climate change progresses, extreme climatic events, including droughts, are expected to increase over the next decade [70]. These are likely to combine the effects of water stress with increased overall heat, which should further impair agronomic performance [71]. Limited water supplies [72] mean that the response of the agricultural sector to these overall drier conditions will require some form of irrigation management to optimise overall water use. Calculating the water requirement of a plant is intricate, as they depend on the species, soil type, and stage of the growth cycle [73]. To track and quantify the combined impact of water and heat stress, we developed the comprehensive and robust SID that can be rapidly determined at different plant developmental stages. In contrast to the SID, traditional stress evaluation indexes such as the Crop Water Stress Index [74,75] require manual and complex measurements. Modern equivalents also tend to focus mainly on water stress levels [76]. In this paper, we successfully demonstrated that the SID was able to rank our eight different environmental conditions. Additionally, this index was able to correctly integrate the impact of the greater heat stress in 2022, underscoring the importance of considering both aspects of heat and water stress. Additionally, the SID was designed to incorporate as few traits as possible and was constructed using traits that can be obtained relatively easily from public meteorological databases or irrigation management tools.

While obtaining the SID is straightforward, its accuracy could be improved by integrating measured soil water deficit through tensiometry data. Furthermore, canopy temperature sensors would greatly improve our results, as our current SID does not account for the effects of canopy cover and shading. Due to the nature of the required data to calculate the SID, it's integration into irrigation decision support tools would be easily feasible. Most commercially available tools already require access to a nearby Météo France or personal agricultural weather station, which routinely measures the data required to calculate the SID. Integrating the SID into such tools would permit farmers to more accurately control their irrigation tools, allowing them to irrigate plants based on their stress level. This environmental classification also allowed us to observe that qualitative traits such as DM digestibility do not respond in a linear fashion to stress severity. The combined impact of water and heat stress in WD 2022 led to a significant decrease in DM digestibility compared to the moderate to severe stress conditions. This result contradicts existing literature, which reports increased grass DM and CW digestibility under water deficit [6,41,45,77,78]. While moderate to severe stress conditions have previously compensated for yield reductions through an increase in DM digestibility through digestible yield, such compensations did not occur under the severe conditions experienced in 2022. This drop in DM digestibility cannot be attributed to a drop in CW digestibility, as it increased under all stress conditions, even under severe stress like that experienced in 2022. In these intense stress situations, the very significant loss of grain production was not compensated for by an increase in CW digestibility and could explain the drop in DM digestibility.

### 4.2. Under severe stress, *p*-hydroxycinnamic acids contents decrease while lignin level remains stable

Several studies across various species have demonstrated that water stress leads to a decrease in lignin content [6,77,79–81], as well as in PCest content [6]. Furthermore, although studies focusing on the isolated effect of heat stress are rarer, they suggest an influence of heat on the lignification process [82]. The synthesis of phenolic compounds represents a significant and irreversible investment of carbon and energy for the plant, subject to various regulatory levels. It is believed that there is a decrease in the allocation of carbon to phenolic compounds in favour of synthesizing soluble compounds, which may participate in osmoregulation and storage to facilitate recovery when water stress stops [81]. It is also commonly asserted that the lignin synthesized under stress have modified structure and composition. Cesarino 2019 [83] pointed out the highly variable response to biotic and abiotic stresses, indicating that no definitive conclusions can be drawn regarding the specific role of individual lignin monomers in response to stress. Our study revealed that irrespective of the stress intensity, the composition and structure of the lignin remained unchanged. Furthermore, we demonstrated

that through the selection for yield and digestibility, breeders indirectly stabilized lignin content in modern hybrids, which is strongly constrained in terms of both inter-genotypic variations and inter-environmental variations. Lignin plays an important structural role in protecting plants against pest attacks, maintaining overall plant health and providing support to vascular tissues, particularly to withstand turgor pressure [84]. The increasing stresses had an impact on phenolic compounds composition, not by reducing lignin content but by sharply lowering hydroxycinnamic acid contents, primarily impacting the synthesis of PCest, secondarily affecting Feest and Feeth. Ferulic and *p*-coumaric acids serve as intermediates in the lignin biosynthetic pathway [85]. The regulation put in place in maize hybrids under varying degrees of water and heat stress is therefore quite remarkable in terms of ensuring that lignin content is maintained and that the drop in C flux is passed on to *p*-hydroxycinnamic acids.

### 4.3. Both biochemical and histological variables significantly explain CW digestibility with different stress response thresholds

Through our multi-trait linear regression analysis, we identified that, under all stress intensity levels, the same types of biochemical traits returned: *p*-hydroxycinnamic acids (particularly PCest) and lignin structural sub-units. In both humid and severe stress conditions, these traits alone effectively explained CW digestibility, with a few histological traits providing additional, minor contributions. These biochemical results echo findings from previous studies conducted on humid conditions [4,86–88]. Surprisingly, under moderate to severe stress conditions, this trend was entirely reversed with histological traits becoming the primary contributors to explaining CW digestibility. Interestingly, the inclusion of histological traits with greater importance into the model was non-redundant with biochemical traits, as the same trait types observed earlier were also present in these conditions, though with less importance. However, this trend did not persist under the severe stress condition, where histology once again took a secondary role to biochemical traits.

Given that future environmental conditions are likely to involve a combination of moderate to severe stress, it is essential to further investigate the impact of these combined stress conditions on future forage quality research. Such studies will need to combine both biochemical and histological approaches, as our findings demonstrated a significant non-redundancy between these types of traits depending on the environmental context. The delayed response of histological traits to stress levels is consistent with previous findings indicating that biochemical traits in maize [6,35,45] and other grasses [80,81] exhibit earlier response to lower stress levels, whereas histological traits tend to respond only under pronounced stress intensities [34,89].

The different histological tissue types associated to CW digestibility variations were also affected by stress intensity, with DRT and LRT showing differential responses under the 2022 environmental conditions. The combined stresses during that year likely unveiled histological disparities, as previously mentioned. Changes in composition between DRT and LRT, and not only between the overall rind and pith, could also explain the variations observed in 2022. Correlation analysis identified that, in humid conditions, while DRT was often positively correlated with PCest and β04 yield, LRT was negatively correlated to these traits. This order of correlation changes in WD1 2022 and WD 2022, with LRT now being positively associated with these highly indigestible CW elements. With increasing load intensity, denser elements would preferentially deposit in the LRT, closer to the pith, than in the DRT, which is closer to the epidermis. Since the LRT-to-rind ratio was much smaller than the DRT ratio under WD 2022, a highly indigestible LRT ring would form under severe stress conditions, which would be denser than those found at other stress levels. Plants grown under severe stress conditions also exhibited a blue ring, a phenomenon previously observed only in inbred maize lines [5,6,40]. While such bands had been previously observed at a smaller scale in hybrids, those grown in WD 2022 are the first commercial hybrids in our studies to replicate results found in inbred lines. This blue ring is also presented, though greatly reduced, in humid conditions. The use of the new FASGA image segmentation plugin [58] has also permitted the discovery of a smaller red ring in these humid conditions. This red ring appears adjacent to the space between the nonlignified cortical parenchyma cells and the rind, progressively becoming more prominent under humid conditions. This thin layer of lignified cells is not

actually present in the rind and appears to be a form of parenchyma. It is conceivable that under these conditions, the overall plant height and the ear weight (located just above this internode) promote a lignification pattern with a secondary layer of densely lignified tissues in the pith just before the rind. This could act as an internal structural support layer to assist the rind. Given that the rind is a denser tissue that is more prone to breakage, this more malleable dense-pith ring could serve as a more elastic support link. Its absence in the individuals of WD 2022 (and in all conditions in 2021) could be explained by this greater amount of denser DRT, and the overall lack or reduction in ear weight.

These findings led to the elaboration of a trait response model (Fig 5). In this model, the role of histological traits in the determination of CW digestibility rises under moderate to severe stress conditions. This increased importance of histological traits is not observed under severe stress conditions, where digestible yield reaches its lowest levels. We propose that, while lignin content must be maintained at current levels, future selection efforts should concentrate on a duet of trait types: biochemical, with *p*-hydroxycinnamic acids and lignin structure, and histological, with both rind and pith tissues. Targeting both traits will be necessary to guarantee further gains in CW digestibility. Under moderate to severe stress conditions, histological traits contribute as much or even more to CW digestibility than biochemical traits. Additionally, we failed to identify any negative correlations between histological and agronomic traits.

## 5. Conclusion

Through precise control of different water stress conditions and assessment via the SID, we gained insights into the impact of increased stress levels on both the yield and quality of forage maize. The SID we proposed could be used in real-time to trigger irrigation according to set stress thresholds and plant development stages. This tool will be invaluable for avoiding stress around the anthesis and silking period, thus favouring ear establishment by triggering irrigation, and could also be used at different time points to allow controlled stress and enhance CW digestibility. The SID would

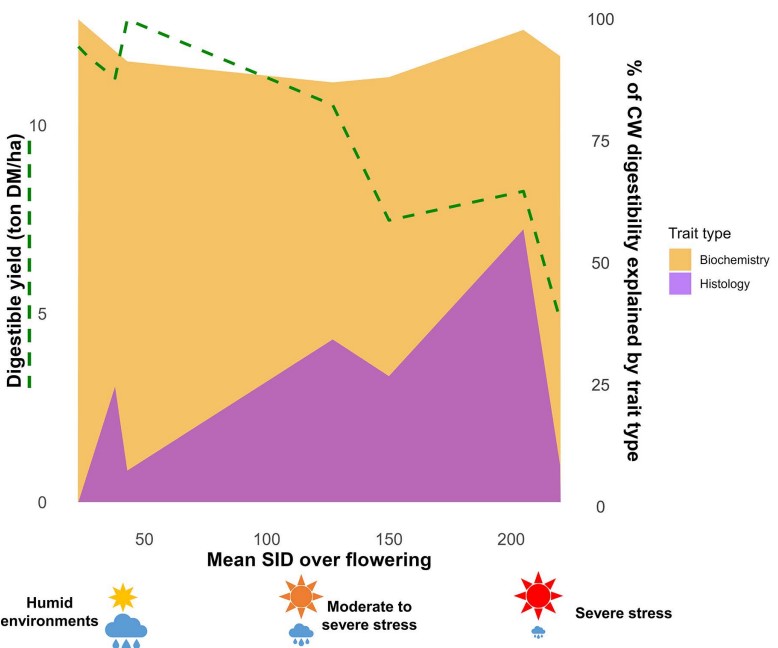

**Fig 5. Model of forage maize digestibility response under progressively increasing water and heat stress intensity.** Digestible yield is represented at the mean level for the different conditions. The proportion of the different trait types explaining CW digestibility are graphically represented here with values obtained from the different regression models, dependent on the given condition.

therefore be a powerful tool for managing improvements in digestible yield. Beyond that, our study underscores the importance of prioritizing enhancements in CW digestibility under stress conditions to maintain or enhance DM digestibility, ultimately affecting the energy content of forage maize. This can be achieved through the integration of new biochemical and histological selection traits. We have demonstrated that lignin content remains nearly constant in maize hybrids, making it less viable as a target for digestibility improvement. Instead, decreasing *p*-coumaric acid content and increasing the proportion of digestible pith tissues appear to be the most promising strategies for enhancing digestibility. Additionally, our findings revealed the regulatory mechanisms at play in maize hybrids under severe water and heat stress, ensuring the maintenance of lignin content, while channelling the decrease in carbon flux toward *p*-hydroxycinnamic acids. Finally, our study unveils that moderate water and heat stress serve as an ideal environmental condition for improving digestible yield, with both biochemical and histological traits exerting similar influence.

## Supporting information

**S1 Table. Main climatic and maintenance data related to the eight different conditions studied.** [1]Total quantity from sowing to harvest; [2]Average temperature or humidity from sowing to harvest; [3]Estimated Penman evapotranspiration; [4]Modified growing degree days (6–30 base) from sowing to harvest.
(XLSX)

**S2 Table. Hybrids used in various growing conditions at Mauguio and Versailles during 2021 and 2022.** MGDD data was obtained from the different breeding companies.
(XLSX)

**S3 Table. List of traits introduced into the regression analysis.**
(XLSX)

**S4 Table. ANOVA analysis of all traits studied across conditions is provided, along with the model and significance levels.** C: condition; c: column; r: row; V: variety; VC: interaction between V and C. Significance levels: $P > 0.05$ NS; $P < 0.1$.; $P < 0.05$ *; $P < 0.01$ **; $P < 0.001$ ***.
(XLSX)

**S1 Fig. Impact of stress intensity across 2021 and 2022 on the ratios of rind tissues.** (A) DRT to rind and (B) LRT to rind ratios to the SID value measured across anthesis and silking (Pflo) per variety under the eight conditions. Letters on top of bars represent the Tukey class for the given condition (P < 0.05).
(TIF)

## Author contributions

**Conceptualization:** Oscar Main, Florence Meunier, Marie-Pierre Jacquemot, Nathalie Mangel, Anthony Uijttewaal, Matthieu Reymond, Sylvie Coursol, Valérie Méchin.

**Data curation:** Oscar Main, Matthieu Reymond, Sylvie Coursol, Valérie Méchin.

**Formal analysis:** Oscar Main, Ana López-Malvar, Nathalie Mangel, Matthieu Reymond, Sylvie Coursol, Valérie Méchin.

**Funding acquisition:** Nathalie Mangel, Anthony Uijttewaal, Sylvie Coursol, Valérie Méchin.

**Investigation:** Oscar Main, Florence Meunier, Sophie Guillaume, Marie-Pierre Jacquemot, Paul-Louis Lopez-Marnet, Nathalie Mangel, Matthieu Reymond, Sylvie Coursol, Valérie Méchin.

**Methodology:** Oscar Main, Ana López-Malvar, Florence Meunier, Sophie Guillaume, Marie-Pierre Jacquemot, Paul-Louis Lopez-Marnet, Charlène Barotin, Anne Marmagne, Laurent Cézard, Sébastien Fargier, Sébastien Rey, Nathalie Mangel, Matthieu Reymond, Valérie Méchin.

**Project administration:** Florence Meunier, Nathalie Mangel, Anthony Uijttewaal, Sylvie Coursol, Valérie Méchin.

**Resources:** Oscar Main, Pierre Larsonneau, Anthony Uijttewaal, Matthieu Reymond, Sylvie Coursol, Valérie Méchin.

**Software:** Oscar Main, Paul-Louis Lopez-Marnet, Pierre Larsonneau, Anthony Uijttewaal, Matthieu Reymond, Valérie Méchin.

**Supervision:** Oscar Main, Florence Meunier, Nathalie Mangel, Anthony Uijttewaal, Matthieu Reymond, Sylvie Coursol, Valérie Méchin.

**Validation:** Oscar Main, Ana López-Malvar, Nathalie Mangel, Anthony Uijttewaal, Matthieu Reymond, Sylvie Coursol, Valérie Méchin.

**Visualization:** Oscar Main, Matthieu Reymond, Valérie Méchin.

**Writing – original draft:** Oscar Main, Ana López-Malvar, Nathalie Mangel, Anthony Uijttewaal, Matthieu Reymond, Sylvie Coursol, Valérie Méchin.

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
