## [Decision Letter · Decision Letter 0]

16 Jul 2025

Dear Dr. Main,

Thank you for submitting your manuscript to PLOS ONE. After careful consideration, we feel that it has merit but does not fully meet PLOS ONE’s publication criteria as it currently stands. Therefore, we invite you to submit a revised version of the manuscript that addresses the points raised during the review process.

**Both reviewers considered that your work is interesting and certainly contributes to our understanding of the molecular mechanisms underlying biomass recalcitrance and the effects of abiotic stresses on cell wall composition/formation and, consequently, on biomass digestibility. Nevertheless, they made several suggestions for your consideration. Please, note that reviewer #1 was more strict and also suggested additional experiments, which you might consider as relevant. **

We look forward to receiving your revised manuscript.

Kind regards,

Igor Cesarino, Ph.D

Academic Editor

PLOS ONE

**Journal Requirements:**

1. When submitting your revision, we need you to address these additional requirements. Please ensure that your manuscript meets PLOS ONE's style requirements, including those for file naming. The PLOS ONE style templates can be found at https://journals.plos.org/plosone/s/file?id=wjVg/PLOSOne_formatting_sample_main_body.pdf and https://journals.plos.org/plosone/s/file?id=ba62/PLOSOne_formatting_sample_title_authors_affiliations.pdf 2. Thank you for stating in your Funding Statement: This work is supported by the Plant2Pro Carnot institute in the frame of the MAMMA MIA project. Plant2Pro is supported by ANR (agreement #20 CARN 0024 01). This work has benefited from the support of the IJPB’s Plant Observatory platform PO-Cyto. The IJPB laboratory also benefits from the support of the Saclay Plant Sciences-SPS (ANR-17-EUR-0007).  Please provide an amended statement that declares *all* the funding or sources of support (whether external or internal to your organization) received during this study, as detailed online in our guide for authors at http://journals.plos.org/plosone/s/submit-now.  Please also include the statement “There was no additional external funding received for this study.” in your updated Funding Statement. Please include your amended Funding Statement within your cover letter. We will change the online submission form on your behalf. 3. We note that you have indicated that there are restrictions to data sharing for this study. PLOS only allows data to be available upon request if there are legal or ethical restrictions on sharing data publicly. For more information on unacceptable data access restrictions, please see http://journals.plos.org/plosone/s/data-availability#loc-unacceptable-data-access-restrictions.  Before we proceed with your manuscript, please address the following prompts: a) If there are ethical or legal restrictions on sharing a de-identified data set, please explain them in detail (e.g., data contain potentially identifying or sensitive patient information, data are owned by a third-party organization, etc.) and who has imposed them (e.g., a Research Ethics Committee or Institutional Review Board, etc.). Please also provide contact information for a data access committee, ethics committee, or other institutional body to which data requests may be sent. b) If there are no restrictions, please upload the minimal anonymized data set necessary to replicate your study findings to a stable, public repository and provide us with the relevant URLs, DOIs, or accession numbers. For a list of recommended repositories, please seehttps://journals.plos.org/plosone/s/recommended-repositories. You also have the option of uploading the data as Supporting Information files, but we would recommend depositing data directly to a data repository if possible. We will update your Data Availability statement on your behalf to reflect the information you provide. 4. We note that you have included the phrase “data not shown” in your manuscript. Unfortunately, this does not meet our data sharing requirements. PLOS does not permit references to inaccessible data. We require that authors provide all relevant data within the paper, Supporting Information files, or in an acceptable, public repository. Please add a citation to support this phrase or upload the data that corresponds with these findings to a stable repository (such as Figshare or Dryad) and provide and URLs, DOIs, or accession numbers that may be used to access these data. Or, if the data are not a core part of the research being presented in your study, we ask that you remove the phrase that refers to these data. 5. If the reviewer comments include a recommendation to cite specific previously published works, please review and evaluate these publications to determine whether they are relevant and should be cited. There is no requirement to cite these works unless the editor has indicated otherwise. 

Reviewers' comments:

**Comments to the Author**

1. Is the manuscript technically sound, and do the data support the conclusions?

Reviewer #1: Yes

Reviewer #2: Yes

2. Has the statistical analysis been performed appropriately and rigorously?

Reviewer #1: Yes

Reviewer #2: Yes

3. Have the authors made all data underlying the findings in their manuscript fully available?

Reviewer #1: No

Reviewer #2: Yes

4. Is the manuscript presented in an intelligible fashion and written in standard English?

Reviewer #1: Yes

Reviewer #2: Yes

**Reviewer #1:**  The authors conducted a very interesting research into the relationship between drought stress and cell wall (CW) digestibility in maize forage varieties. I would like to congratulate the authors on performing drought treatment on maize plants in the field despite the associated climate-related issues. One of the most interesting aspects of the manuscript is the development of an environmental stress index (SID) that helped the authors determine the severity of the stress applied to different maize varieties in the field. The authors also attempted to correlate the severity of the stress with changes in CW biochemistry and histological features that could explain the observed changes in digestibility. Finally authors try to describe several biochemical and histological features that csan be used as markers of digestibility in the global warming context that we wil suffer in the future.

Although I think the work successfully identifies biochemical and histological markers that can be used to evaluate the digestibility of maize varieties in the field, several points in the manuscript should be explained more clearly. A more detailed analysis of CW components would also help to explain the changes in CW digestibility due to drought stress that the authors have demonstrated.

Here I will try to summarize the major points:

One of my main concerns relates to the absence of SID differences between the moderate and severe treatments in the two years studied. The authors highlight that the only period in which the two treatments have a different SID index is between 01/07 and 15/07 (the 2022 treatment had a higher index), which corresponds to the flowering period of the female and male plants. Although the authors state that this period is optimal because it covers the most important developmental stages of maize growth, and that maize is more sensitive to stresses during this time, I believe this needs further explanation. In my view, during flowering, the maize stover has mostly developed, and the changes that severe (2022) drought stress can make to this structure are not substantial enough to explain the changes in CW digestibility. Considering that the SID severity in 2022 was only higher during the flowering period, I suppose these results would correlate with the lower ear and grain productivity in these treatments, as demonstrated by the authors. Therefore, I think the starch content (an indicator of the proportions of ears and grains in each sample) can explain the observed changes in digestibility (higher in the "moderate" treatments in 2021 and lower in the "severe" treatment in 2022, due to the lower proportion of grains in the latter samples). Although the authors mentioned and discussed this feature in the discussion section, I would recommend including the starch quantifications and further discussing the importance of this grain contribution to digestibility, because, in my opinion, this is a central point of the manuscript.

Another interesting point is that the authors study the biochemical composition of cell walls to explain changes in digestibility, focusing on hydroxycinnamate and lignin composition. It is already known that these components affect the availability of polysaccharides for degradation by digestive enzymes. I think information about polysaccharides, which are components of CW subject to degradation, is important for explaining CW digestibility. Therefore, evaluating at least cellulose content and the amounts of non-cellulosic polysaccharides (especially hemicelluloses which are related to hydroxycinnamates) would provide a broader view of the CW and could help explain differences in digestibility.

The final major point relates to the statistically significant reduction in lignin content, which correlates precisely with the severity of stress. Surprisingly, however, the authors considered the lignin content to be unchanged. The authors consider that, even though the differences in lignin content are significant, they are below the margin of error of the quantification method. However, they did not provide any references describing this feature. I would recommend including references that support this, as I thought the acetyl bromide method was more accurate for quantifying lignin in grass cell walls. I would also recommend that the authors provide a more detailed explanation of why they consider this reduction in lignin to be insufficient to be considered a key factor in CW digestibility, and whether there is more research to support their decision.

There are also some minor points:

-Personally, I am not very familiar with FASGA staining. Therefore, based on the colours alone, I would say that blue tissues are less digestible than pink ones. If I am correct, I would include this information in the Materials and Methods section to help non-experts understand.

I would suggest that the authors avoid using the Saxon genitive, which was sometimes used in the introduction section.

In Table I, the value for the G subunit for WW 2022 seems to be incorrect (332.790.53); please correct this.

- I was unable to find an explanation for concepts such as B04 yield. If an explanation is not provided, it should be included in the text and in the abbreviation section. Similarly, abbreviations such as %ICS and S-PC should also be included in this section.

**Reviewer #2:**  Review of Manuscript Entitled: Targeting Enhanced Digestibility: Prioritizing low pith lignification to complement low p=coumaric acid content and environmental stress intensity increases. This manuscript aims to understand the differences in lignin and lignin-associated traits in hybrid silage maize lines as they undergo drought stress tolerance. This is really interesting work and important in understanding the complex nature of the digestibility of maize silage with all the implications of better feed conversion and corresponding climate change implications, along with breeding for a changing climate, and in particular drought tolerance and the corresponding changes in the components of the lignin pathway.

There is a lot of data to relay in this manuscript to sort through, and I encourage the authors to be tighten up the wording to some degree. The Materials and Methods section contains parts that read like results and maybe be better served to be moved to the results section for succinctness.

A few notes:

1. Page 2: Abstract: please indicate how many lines were tested.

2. Page 6: M&M, can you describe the hybrids any further? Any information on the level of lignin each of these offer? Are they brown midrib mutant hybrids?

3. Page 6: As a general comment, some of the Materials and Methods is a little verbose, and there are some passages that are more results or discussion oriented. It would read easier if this was corrected. For example, the description of what happened in Versailles and Mauguoi sites reads more like a results passage then Materials and Methods section.

4. Page 6: Agronomic analysis: Please refer to flowering dates as days to anthesis and days to silking.

5. Page 10: Statistical analysis: ‘data management and wrangling…’ while we all know what this means perhaps another term other than ‘wrangling’ is more appropriate.

6. Page 10: ‘Due to environmental analysis….we decided to analyse condiitons….’ Please reword to be less conversational

7. Page 13: Table 1 This is a really interesting table. A) I would ask to add days to anthesis, days to silking and the anthesis-silk interval to this table because I want to know how this compares to the other traits, B) Also please add the S:G monolignol ratio to this table for same reason, C) Does the ASI affect the starch/soluble sugar contents due to low pollination?

8. Page 14: 3rd paragraph: ‘DM digestibility is the result of the overall digestibility of these three combined compartments’. Please list the compartments instead.

9. It would be a good idea to include a supplemental figure with the lignin pathway to show where the flex points for coumaric acid and hydroxycinnamic acid are.

10. Do you think that brown midrib hybrids would react in this same manner?

**Do you want your identity to be public for this peer review?** For information about this choice, including consent withdrawal, please see our Privacy Policy

Reviewer #1: **Yes: ** Asier Largo Gosens

Reviewer #2: No

---

## [Author Response · Author response to Decision Letter 1]

6 Oct 2025

Dear editor and reviewers,

Please find enclosed the revised version of the manuscript entitled "Targeting Enhanced Digestibility: Prioritizing Low Pith Lignification to Complement low p-Coumaric Acid content as environmental stress intensity increases" submitted to PlosOne.

As requested by the editor, I’ve attached below the amended funding and data availability paragraphs. Regarding the funding statement, our institute requires that the last two phrases in the paragraph (concerning the PO-Cyto platform and the SPS support) remain unchanged. However, this work received no funding from either of these partners but rather benefited from the overall funding which was used to allow day-to-day experiments throughout the institute. I have added a phrase to try and explain this difference but am fully open to alternatives if you can recommend a better way to express this.

Funding

This work was entirely funded by the Plant2Pro Carnot institute in the frame of the MAMMA MIA project. Plant2Pro is supported by ANR (agreement #20 CARN 0024 01). This work has benefited from the support of IJPB's Plant Observatory platforms PO-Cyto (imaging and microscopy plateform). The IJPB laboratory also benefits from the support of the Saclay Plant Sciences-SPS (ANR-17-EUR-0007). The funders had no role in study design, data collection and analysis, decision to publish, or preparation of the manuscript. There was no additional external funding received for this study.

Data availability statement

All raw data, alongside several calculated variables, are freely available in two main databases attached to this publication. The first contains the agronomic and biochemical data, while the second contains histological data. Both can be accessed through https://zenodo.org/records/17199793.

We have addressed responses and changes to all reviewers’ suggestions,

Below you can find a point-by-point response to your comments. For ease of reading, we have colored the original comments in blue and left our responses in black.

Reviewer #1: The authors conducted a very interesting research into the relationship between drought stress and cell wall (CW) digestibility in maize forage varieties. I would like to congratulate the authors on performing drought treatment on maize plants in the field despite the associated climate-related issues. One of the most interesting aspects of the manuscript is the development of an environmental stress index (SID) that helped the authors determine the severity of the stress applied to different maize varieties in the field. The authors also attempted to correlate the severity of the stress with changes in CW biochemistry and histological features that could explain the observed changes in digestibility. Finally authors try to describe several biochemical and histological features that csan be used as markers of digestibility in the global warming context that we wil suffer in the future.

Although I think the work successfully identifies biochemical and histological markers that can be used to evaluate the digestibility of maize varieties in the field, several points in the manuscript should be explained more clearly. A more detailed analysis of CW components would also help to explain the changes in CW digestibility due to drought stress that the authors have demonstrated.

Here I will try to summarize the major points:

One of my main concerns relates to the absence of SID differences between the moderate and severe treatments in the two years studied. The authors highlight that the only period in which the two treatments have a different SID index is between 01/07 and 15/07 (the 2022 treatment had a higher index), which corresponds to the flowering period of the female and male plants. Although the authors state that this period is optimal because it covers the most important developmental stages of maize growth, and that maize is more sensitive to stresses during this time, I believe this needs further explanation. In my view, during flowering, the maize stover has mostly developed, and the changes that severe (2022) drought stress can make to this structure are not substantial enough to explain the changes in CW digestibility. Considering that the SID severity in 2022 was only higher during the flowering period, I suppose these results would correlate with the lower ear and grain productivity in these treatments, as demonstrated by the authors. Therefore, I think the starch content (an indicator of the proportions of ears and grains in each sample) can explain the observed changes in digestibility (higher in the "moderate" treatments in 2021 and lower in the "severe" treatment in 2022, due to the lower proportion of grains in the latter samples). Although the authors mentioned and discussed this feature in the discussion section, I would recommend including the starch quantifications and further discussing the importance of this grain contribution to digestibility, because, in my opinion, this is a central point of the manuscript.

Previous research within the team on inbred maize lines (Zhang et al., 2011) has shown that the deposition of secondary cell wall components is greatly concentrated around silking. Effectively, elements such as lignin are exponentially deposited in and around this physiological stage, after which their deposition reaches a point of stagnation. In our study, 82% of our biological material silked between the 1st and 15th of July, with the mean reached on the 11th. A few individuals flowered after these dates but remain an absolute minority. If we assume by extension that Zhang’s results also apply to hybrids, the dates selected for the stress index here covers the periods where secondary cell wall element deposition reaches a maximum. Additionally, water stress has been applied well before this stage and our experimental program expects stress levels to be well implemented over this key period, which was the case in our study, and is therefore likely to have a strong impact on secondary cell wall composition.

Regarding starch content, this variable is effectively very important in forage maize selection and evaluation. Previous research has indicated that starch is almost entirely degraded throughout the time spent in the rumen and has been found to not be a significant factor impacting dry matter digestibility (Peyrat, 2014). Our research really intends to concentrate upon secondary cell wall digestibility however, which is unaffected by starch content as most protocols require a de-starching process before digestibility can occur. Therefore, we are more interested in the phenolic compounds which are known for impacting secondary cell wall digestibility. Additionally, though we did not perform laboratory-measurements of starch content, NIRS-predicted values obtained within our dataset did not reveal any major implication of this trait regarding digestibility values.

Another interesting point is that the authors study the biochemical composition of cell walls to explain changes in digestibility, focusing on hydroxycinnamate and lignin composition. It is already known that these components affect the availability of polysaccharides for degradation by digestive enzymes. I think information about polysaccharides, which are components of CW subject to degradation, is important for explaining CW digestibility. Therefore, evaluating at least cellulose content and the amounts of non-cellulosic polysaccharides (especially hemicelluloses which are related to hydroxycinnamates) would provide a broader view of the CW and could help explain differences in digestibility.

We agree on the importance of the cellulose and hemicellulose fractions in the cell wall. However, in this study we have mainly focused on the structural components of the cell wall. Based on our results, it is not possible to determine whether the observed changes in cellulose/hemicellulose content are due to an actual increase or decrease in polysaccharide abundance under water deficit conditions, to a reduced reinforcement of the cell wall that renders these polymers more accessible and easier to extract, or to a genuine modification in their abundance under stress.

We did however quantify the etherified and esterified ferulic acid contents which do somewhat overlap sugar hemicellulose and cellulose contents, though no major role or impact was identified on either of these traits.

The final major point relates to the statistically significant reduction in lignin content, which correlates precisely with the severity of stress. Surprisingly, however, the authors considered the lignin content to be unchanged. The authors consider that, even though the differences in lignin content are significant, they are below the margin of error of the quantification method. However, they did not provide any references describing this feature. I would recommend including references that support this, as I thought the acetyl bromide method was more accurate for quantifying lignin in grass cell walls. I would also recommend that the authors provide a more detailed explanation of why they consider this reduction in lignin to be insufficient to be considered a key factor in CW digestibility, and whether there is more research to support their decision.

We acknowledge that ABL measurements are very precise, and that the limited variability observed is indeed correlated. However, this range of variation is extremely small and essentially falls within the experimental error. This is not a matter of methodological limitation, but rather of biological stability: lignin content has already been strongly selected for over several generations, resulting in very stable levels in modern hybrids compared to historical lines and hybrids. This highlights the necessity of identifying new traits. We will adapt the text to clarify this point and avoid suggesting a methodological limitation. Moreover, even if a higher variability in lignin content were present, we have often emphasized that lignin should not be the main target, as it has been correlated in different ways, for instance, with grain yield and biotic stress resistance.

There are also some minor points:

-Personally, I am not very familiar with FASGA staining. Therefore, based on the colours alone, I would say that blue tissues are less digestible than pink ones. If I am correct, I would include this information in the Materials and Methods section to help non-experts understand.

As suggested by the reviewer we have included more details regarding FASGA staining and its interpretation, as well as we have updated the references

I would suggest that the authors avoid using the Saxon genitive, which was sometimes used in the introduction section.

The manuscript was initially written by a native English speaker. During revision, we made a conscious effort to limit the use of the Saxon genitive, aiming to improve readability

In Table I, the value for the G subunit for WW 2022 seems to be incorrect (332.790.53); please correct this.

Thank you for the remark, the value has been corrected in the table.

- I was unable to find an explanation for concepts such as B04 yield. If an explanation is not provided, it should be included in the text and in the abbreviation section. Similarly, abbreviations such as %ICS and S-PC should also be included in this section.

Thank you for your remark, we have adapted the text to include further explanations regarding this term in the material and methods section. Additionally, the mentioned abbreviations have been added.

Reviewer #2: Review of Manuscript Entitled: Targeting Enhanced Digestibility: Prioritizing low pith lignification to complement low p=coumaric acid content and environmental stress intensity increases. This manuscript aims to understand the differences in lignin and lignin-associated traits in hybrid silage maize lines as they undergo drought stress tolerance. This is really interesting work and important in understanding the complex nature of the digestibility of maize silage with all the implications of better feed conversion and corresponding climate change implications, along with breeding for a changing climate, and in particular drought tolerance and the corresponding changes in the components of the lignin pathway.

There is a lot of data to relay in this manuscript to sort through, and I encourage the authors to be tighten up the wording to some degree. The Materials and Methods section contains parts that read like results and maybe be better served to be moved to the results section for succinctness.

A few notes:

1. Page 2: Abstract: please indicate how many lines were tested.

We have added the number of lines evaluated in the abstract.

2. Page 6: M&M, can you describe the hybrids any further? Any information on the level of lignin each of these offer? Are they brown midrib mutant hybrids?

All the information provided in M&M is a summary of information available for these hybrids. As the hybrid material is obtained from industry partners, information is limited to what is publicly accessible.

3. Page 6: As a general comment, some of the Materials and Methods is a little verbose, and there are some passages that are more results or discussion oriented. It would read easier if this was corrected. For example, the description of what happened in Versailles and Mauguoi sites reads more like a results passage then Materials and Methods section.

Thank you for your suggestion. Efforts have been made to make the Materials and Methods section less verbose.

4. Page 6: Agronomic analysis: Please refer to flowering dates as days to anthesis and days to silking.

As suggested, the term flowering dated has been changed to days to anthesis and days to silking

5. Page 10: Statistical analysis: ‘data management and wrangling…’ while we all know what this means perhaps another term other than ‘wrangling’ is more appropriate.

The term has been updated following the suggestion.

6. Page 10: ‘Due to environmental analysis….we decided to analyse condiitons….’ Please reword to be less conversational

It has been updated following the suggestion.

7. Page 13: Table 1 This is a really interesting table. A) I would ask to add days to anthesis, days to silking and the anthesis-silk interval to this table because I want to know how this compares to the other traits, B) Also please add the S:G monolignol ratio to this table for same reason, C) Does the ASI affect the starch/soluble sugar contents due to low pollination?

Following your comments, to avoid overloading the present table we have added the indicated information to the overall data file that is now attached to this paper. Regarding the impact of the ASI on the starch or soluble sugar contents, differences were observed between the different environments observed in our work. However, no links were identified between either trait and cell wall digestibility or other phenolic compounds.

8. Page 14: 3rd paragraph: ‘DM digestibility is the result of the overall digestibility of these three combined compartments’. Please list the compartments instead.

Following the review, we have included the list of the combined compartments

9. It would be a good idea to include a supplemental figure with the lignin pathway to show where the flex points for coumaric acid and hydroxycinnamic acid are.

While we appreciate the suggestion, we don’t see the value of adding this supplementary figure. Our work is not based upon a molecular approach wherein we could study the actors responsible for potential flex points in the lignin pathway. Many other papers (Oliveira, et al. (2025). Trends in Biotechnology.; Dixon, & Barros, (2019). Open biology) have studied this question and presented different potential flex points.

10. Do you think that brown midrib hybrids would react in this same manner?

The authors would be curious to study this subject. However, due to the use of hybrids in our tests, this material would be too genetically different from the bmr mutants. Our tests would be incompatible in this regard, and results would likely vary depending on the studied mutants. In the author’s opinions the negative impact of the bmr mutations would be further enhanced unde

---

## [Decision Letter · Decision Letter 1]

18 Nov 2025

Targeting enhanced digestibility: prioritizing low pith lignification to complement low p-coumaric acid content as environmental stress intensity increases

PONE-D-25-24614R1

Dear Dr. Main,

We’re pleased to inform you that your manuscript has been judged scientifically suitable for publication and will be formally accepted for publication once it meets all outstanding technical requirements.

Kind regards,

Igor Cesarino, Ph.D

Academic Editor

PLOS ONE

Additional Editor Comments (optional):

Reviewers' comments:

Reviewer's Responses to Questions

**Comments to the Author**

Reviewer #1: All comments have been addressed

2. Is the manuscript technically sound, and do the data support the conclusions?

Reviewer #1: Yes

3. Has the statistical analysis been performed appropriately and rigorously?

Reviewer #1: Yes

4. Have the authors made all data underlying the findings in their manuscript fully available?

Reviewer #1: Yes

5. Is the manuscript presented in an intelligible fashion and written in standard English?

Reviewer #1: Yes

Reviewer #1: I would like to congratulate the authors on successfully answering all my questions and addressing all my concerns. I believe the manuscript main issues have been addressed, and I recommend its publication on Plos One.

**Do you want your identity to be public for this peer review?** For information about this choice, including consent withdrawal, please see our Privacy Policy

Reviewer #1: **Yes: ** Asier Largo-Gosens

---

## [Editor Report · Acceptance letter]

PONE-D-25-24614R1

PLOS ONE

Dear Dr. Main,

I'm pleased to inform you that your manuscript has been deemed suitable for publication in PLOS ONE. Congratulations! Your manuscript is now being handed over to our production team.

Kind regards,

on behalf of

Dr. Igor Cesarino

Academic Editor

PLOS ONE